# Severe leptospirosis in tropical and non-tropical areas: A comparison of two french, multicentre, retrospective cohorts

**Jérôme Allyn**[1,2]*, **Arnaud-Félix Miailhe**[3], **Benjamin Delmas**[1], **Lucas Marti**[1], **Nicolas Allou**[1,2], **Julien Jabot**[1], **Jean Reignier**[4]

**1** Réanimation Polyvalente, Centre hospitalier universitaire Félix Guyon, La Réunion, Bellepierre Saint-Denis cedex, France, **2** Département d'Informatique Clinique, Centre hospitalier universitaire Félix Guyon, La Réunion, Bellepierre Saint-Denis cedex, France, **3** Nantes Université, CHU Nantes, Médecine Intensive Réanimation, Nantes, France, **4** Nantes Université, CHU Nantes, Médecine Intensive Réanimation, Movement—Interactions—Performance, MIP, UR 4334, Nantes, France

* jerome.allyn@chu-reunion.fr

## Abstract

**Data Availability Statement:** The data used for this publication are considered pseudonymized personal data. As such, the GDPR as well as French data protection law apply to the use of this

### Background

Leptospirosis is an anthropozoonosis that occurs worldwide but is more common in tropical regions. Severe forms may require intensive care unit (ICU) admission. Whether the clinical patterns and outcomes differ between tropical and non-tropical regions with similar health-care systems is unclear. Our objective here was to address this issue by comparing two cohorts of ICU patients with leptospirosis managed in mainland France and in the overseas French department of Réunion, respectively.

### Methodology/Principal findings

We compared two retrospective cohorts of patients admitted to intensive care for severe leptospirosis, one from Reunion Island in the Indian Ocean (tropical climate) and the other from metropolitan France (temperate climate). Chi-square and Student's t tests were used for comparisons. After grouping the two cohorts, we also performed multiple correspondence analysis and hierarchical clustering to search for distinct clinical phenotypes. The Réunion and Metropolitan France cohorts comprised 128 and 160 patients respectively. Compared with the Réunion cohort, the metropolitan cohort had a higher mean age (42.5±14.1 vs. 51.4 ±16.5 years, p<0.001). Severity scores, length of stay and mortality did not differ between the two cohorts. Three phenotypes were identified: hepato-renal leptospirosis (54.5%) characterized by significant hepatic, renal and coagulation failure, with a mortality of 8.3%; moderately severe leptospirosis (38.5%) with less severe organ failure and the lowest mortality rate (1.8%); and very severe leptospirosis (7%) manifested by neurological, respiratory and cardiovascular failure, with a mortality of 30%.

dataset. For any request for access to data, the applicant can contact: Institutional Review Board (IRB 00014135) of SRLF, available at: https://www.srlf.org/article/charte-soumission-dun-projet-recherche. The applicant is informed that access and reuse of this dataset will require obtaining (i) authorization from the Control Authority. Authority responsible for data protection (the CNIL) and (ii) an ethical opinion from the competent French ethics committee.

**Funding:** The author(s) received no specific funding for this work.

**Competing interests:** The authors have declared that no competing interests exist.

## Conclusions/Significance

The outcomes of severe leptospirosis requiring ICU admission did not differ between tropical and temperate regions with similar healthcare access, practices, and resources, despite some differences in patient characteristics. The identification of three different clinical phenotypes may assist in the early diagnosis and management of severe leptospirosis.

### Author summary

This study aims to discern whether there are variations in the clinical manifestations and outcomes of severe Leptospirosis between tropical and non-tropical regions with comparable healthcare infrastructures. Conducted in France, the research scrutinizes the distinctions in patient characteristics, clinical presentations, and prognoses between a tropical area and a temperate one. The analysis reveals minimal discrepancies in patient profiles and clinical patterns across these regions. Additionally, no statistically significant differences were observed regarding the duration of intensive care unit (ICU) or hospital stays, as well as ICU or hospital mortality rates, between the tropical and non-tropical locales. The findings suggest that the clinical features and outcomes of individuals afflicted with severe leptospirosis exhibit substantial uniformity across tropical and temperate regions endowed with similar healthcare resources. This study contributes valuable insights into the understanding of leptospirosis, highlighting the consistency in disease presentation and prognosis irrespective of climatic distinctions, given comparable healthcare capabilities.

## Introduction

Leptospirosis is a worldwide anthropozoonosis responsible for a global public health burden that is considerably heavier in tropical than in temperate regions [1]. The incidence has been estimated at nearly one million cases, with 60 000 deaths, each year [2–4]. However, the true incidence may be far higher, since notification is not mandatory and many cases probably go undiagnosed. Humans are contaminated via occupational, avocational, or recreational exposure to infected animals, their excreta, or contaminated soil or water. Large epidemics occur after periods of heavy rainfall or flooding, which are more common under tropical climates.

The onset is typically acute and the symptoms mild and non-specific, raising diagnostic challenges [5]. Most of these mild cases resolve spontaneously. However, about 10% of patients experience severe acute illness with life-threatening organ failures and bleeding requiring ICU admission. The liver and kidneys are major target organs and the combination of jaundice, kidney failure, and fever has been known since the late nineteenth century as Weil's disease. Thrombocytopenia is common. Severe pulmonary haemorrhagic syndrome and/or central nervous system involvement may develop. Mortality rates in ICU patients with severe leptospirosis have ranged from 4% in north-eastern, tropical Australia to 44.4% and 52% in two other tropical regions, Sri Lanka and India, respectively [6–8]. Differences in healthcare access and resources may contribute to this considerable variability across areas of similar climate. Whether other factors are involved, such as differences in *Leptospira* species or in population characteristics is unclear. Moreover, no large studies have compared patients with severe leptospirosis in tropical vs. temperate regions. Mainland France has a temperate climate with four

seasons and an annual incidence of leptospirosis of about 1/100 000. The overseas French department Réunion is a tropical island located in the Indian Ocean with a hot rainy season from December to March and a cooler dry season the rest of the year. The incidence of leptospirosis is about 50 to 100 times higher than in mainland France [9]. Severe cases occur in both mainland France and Réunion [10,11]. Healthcare practices and resources are alike in the two regions, which share the same easy access to care via a statutory health insurance system. Equipment and staffing patterns in ICUs are similar, the same professional societies provide guidance, and the same recommendations are applied. Thus, any differences in patient features and/or outcomes would not be ascribable to differences in healthcare.

The primary objective of this study was to compare the characteristics and outcomes of patients admitted to the ICU for severe leptospirosis in a temperate region vs. a tropical region sharing the same healthcare system. To this end, we compared two retrospective cohorts of patients from mainland France and Réunion. The secondary objective was to determine whether distinct clinical phenotypes of severe leptospirosis could be identified.

## Materials and methods

### Ethics approval and consent to participate

This is an ancillary study of two previous studies coordinated by the Félix Guyon university hospital in Réunion and by the university hospital in Nantes in mainland France, respectively. The Félix Guyon institutional review board approved the study in Réunion (#R15009) and the ethics committee of the French Intensive Care Society the study in mainland France (#CE SRLF16-06). In both cases, the requirement for informed consent was waived, in compliance with French law on retrospective healthcare studies of de-identified data.

### Study design

We designed an ancillary study of two previously published, observational, retrospective, cohort studies coordinated by the Réunion university hospital and the Nantes university hospital (mainland France), respectively [10,11]. We retrospectively compared the data collected for these two studies. The current study is reported in accordance with STrengthening the Reporting of OBservational studies in Epidemiology (STROBE) recommendations [12].

The study done in Réunion included consecutive patients admitted for severe leptospirosis to either of the two ICUs on the island between January 2004 and January 2015. The study done in mainland France (LEPTOREA) included consecutive adults admitted for severe leptospirosis to any of the 79 participating ICUs between January 2012 and September 2016 [11].

The authors of the two studies provided the data for the current study and the definitions of the collected variables. We compared only those variables for which the definitions in the original studies were identical. The databases were then merged, with each patient being labelled as belonging to one or the other cohort, and analyses were performed on the overall population to look for distinct clinical phenotypes.

### Definitions

Severe leptospirosis was defined as leptospirosis necessitating ICU admission [13–15]. Documentation by at least one laboratory test was required to define leptospirosis. In Réunion, the tests used were polymerase chain reaction (PCR) on blood and/or urine samples and/or serological testing using the microscopic agglutination test (MAT) and/or enzyme-linked immunosorbent assay. In mainland France, the same tests were used, as well as dark-field microscopy of blood samples. Acute respiratory distress syndrome (ARDS) was defined

according to the Berlin criteria [16]. More details about definitions are available in the additional file.

## Statistical analysis

Categorical variables were described as number (percentage) and continuous variables as mean±standard deviation. Comparisons were with the chi-square test for categorical variables and Student's *t* test for continuous variables.

We sought to identify distinct disease phenotypes by performing multiple correspondence analysis (MCA) followed by hierarchical ascending classification (HAC), using the R program, version 3.2.2 (The R Foundation for Statistical Computing; Vienna, Austria), with the Facto-MineR and explor packages [17,18]. MCA is a form of vector analysis in which a set of individuals is described via a set of categorical variables. The pattern of relationships among the variables can then be represented in a two-dimensional space whose axes (dimensions) are calculated so that most of the variations are concentrated on the first axes, in order to identify groups with common profiles [19]. We determined the cumulative inertia for n = x dimensions, corresponding to the variance in the cohort, and the numbers representing the relationship between two dimensions (e.g., first and second or first and third) for the variables and for the patients. HAC partitions the population into sub-groups via a distance matrix, thereby allowing the construction of a dendrogram. For this analysis, the only variables used were organ failures (e.g., cardiovascular, renal, hepatic), categorised as present or absent, with present being defined as at least two points on the relevant Sequential Organ Failure Assessment sub-score [13].

The cohort study from mainland France included MCA, which used the following variables: age, sex, smoking status, alcohol abuse, organ failures (SOFA sub-scores at ICU admission), and clinical findings at ICU admission (fever, myalgia, arthralgia, headache, delirium, impaired consciousness, vomiting, diarrhoea, abdominal pain, dyspnoea, cough, and bleeding [including gastrointestinal bleeding, epistaxis, purpura, and haematuria]) [11,13]. For the current study, to avoid bias related to differences in definitions between the two previous studies, MCA was based only on organ failures as defined above. The inclusion cohort and ICU death were coded as illustrative variables.

## Results

The current study included 288 adults admitted to the ICU for severe leptospirosis, 128 in Réunion and 160 in mainland France. The number of patients per ICU and per year was 6 in Réunion compared to only 0.4 in mainland France. Cases were more common between March and May in Réunion and between August and October in mainland France (S1 Fig).

### Patient characteristics and clinical presentation on ICU admission

Mean age was significantly lower in the Réunion cohort than in the mainland cohort (Table 1). Comorbidities and time from symptom onset to hospital admission were not different. Myalgia, jaundice, and haemoptysis were more common in Réunion, where the mean creatine kinase and bilirubin levels were higher and the mean prothrombin time lower. There were significantly more patients with cardiovascular failure according to SOFA sub score in the mainland cohort than in Réunion (36.9%, versus 18.8% respectively, p<0.001). However, the severity scores (total SOFA score and Simplified Acute Physiology Score version II [SAPSII]) did not differ significantly between the two groups [13,20].

**Table 1. Main features of the study patients on ICU admission.**

| | Total N = 288 | Mainland France N = 160 | Réunion N = 128 | p value |
|---|---|---|---|---|
| Age, years, mean±SD | 47 ± 16 | 51 ± 17 | 43 ± 14 | <0.001 |
| Male, n (%) | 265 (92) | 146 (91.2) | 119 (93) | 0.67 |
| Smoking, n (%) | 94 (32.6) | 49 (30.6) | 45 (35.2) | 0.23 |
| Chronic alcohol abuse, n (%) | 63 (21.9) | 29 (18.1) | 34 (26.6) | 0.16 |
| **Comorbidities, n (%)** | | | | |
| Diabetes mellitus | 23 (8.0) | 9 (5.6) | 14 (10.9) | 0.13 |
| Liver cirrhosis | 1 (0.3) | 4 (3) | 1 (0.8) | 0.44 |
| Cancer or immune deficiency | 2 (0.7) | 2 (1.2) | 0 (0) | 0.50 |
| Chronic kidney disease | 1 (0.3) | 0 | 0 | 1 |
| Chronic respiratory failure | 0 (0) | 0 (0) | 0 (0) | - |
| **Severity scores on admission, mean±SD** | | | | |
| SAPS II[a] | 43 ± 21 | 44 ± 22 | 40 ± 19 | 0.12 |
| SOFA score[b] | 10 ± 4 | 10 ± 4 | 10 ± 4 | 0.83 |
| **SOFA sub-score ≥2[a], n (%)** | | | | |
| Central nervous system failure | 20 (6.9) | 8 (5) | 12 (9.4) | 0.22 |
| Renal failure | 221 (76.7) | 121 (75.6) | 100 (78.1) | 1 |
| Cardiovascular failure | 83 (28.8) | 59 (36.9) | 24 (18.8) | <0.001 |
| Liver failure | 222 (77.1) | 119 (74.4) | 103 (80.5) | 0.33 |
| Coagulation failure | 238 (82.6) | 133 (83.1) | 105 (82) | 0.76 |
| Respiratory failure | 76 (26.4) | 44 (27.5) | 32 (25) | 0.73 |
| **Time from symptom onset to hospital admission, days, mean±SD** | 4.9 ± 3.1 | 5.1 ± 3.8 | 4.7 ± 1.8 | 0.24 |
| **Antibiotics within 24 hours in ICU, n (%)** | 275 (95.5) | 147 (91.9) | 128 (100) | 0.03 |
| **Symptoms on ICU admission, n (%)** | | | | |
| Fever | 244 (84.7) | 135 (84.4) | 109 (85.2) | 0.5 |
| Myalgia | 197 (68.4) | 95 (59.4) | 102 (79.7) | <0.001 |
| Jaundice | 176 (61.1) | 74 (46.2) | 102 (79.7) | <0.001 |
| Diarrhoea | NA | 43 (26.8) | NA | - |
| Dyspnoea | 76 (26.4) | 42 (26.2) | 34 (26.6) | 1 |
| Abdominal pain | 84 (29.2) | 41 (25.6) | 43 (33.6) | 0.15 |
| Arthralgia | 76 (26.4) | 35 (21.9) | 41 (32) | 0.06 |
| Coughing | 80 (27.8) | 35 (21.9) | 45 (35.2) | 0.02 |
| Meningeal syndrome | 7 (2.4) | 3 (1.9) | 4 (3.1) | 0.71 |
| Haemoptysis | 51 (17.7) | 12 (7.5) | 39 (30.5) | <0.001 |
| **Laboratory data on ICU admission (worst value within 24 h), mean±SD** | | | | |
| Lactate (mmol/L) | 2.3 ± 2.5 | 2.2 ± 1.8 | 2.4 ± 3.2 | 0.68 |
| Bilirubin (µmol/L) | 159 ± 157 | 129 ± 128 | 197 ± 179 | <0.001 |
| Alanine aminotransferase (IU/L) | 145 ± 516 | 175 ± 682 | 108 ± 144 | 0.28 |
| Aspartate aminotransferase (IU/L) | 227 ± 743 | 250 ± 980 | 198 ± 230 | 0.56 |
| Haemoglobin (Giga/dL) | 11.5 ± 2.0 | 11.4 ± 1.8 | 11.5 ± 2.2 | 0.82 |
| Platelets (Giga/L) | 59 ± 48 | 57 ± 45 | 61 ± 51 | 0.57 |
| Leucocytes (Giga/L) | 12.0 ± 7.2 | 11.5 ± 8.0 | 12.7 ± 6.1 | 0.19 |
| Prothrombin time | 78 ± 20 | 81 ± 16 | 75 ± 25 | 0.02 |
| Creatinine (µmol/L) | 354 ± 206 | 346 ± 195 | 365 ± 217 | 0.43 |
| Urea (mmol/L) | 21 ± 12 | 22 ± 13 | 18 ± 10 | 0.005 |
| Potassium (mmol/L) | 3.7 ± 0.7 | 3.8 ± 0.6 | 3.6 ± 0.7 | 0.04 |
| Creatine kinase (IU/L) | 2729 ± 4103 | 1855 ± 2968 | 3645 ± 4875 | 0.002 |

*(Continued)*

**Table 1.** (Continued)

| | Total N = 288 | Mainland France N = 160 | Réunion N = 128 | *p* value |
|---|---|---|---|---|
| C-reactive protein (mg/L) | 233 ± 109 | 236 ± 109 | 229 ± 109 | 0.65 |

SAPS II, Simplified Acute Physiology Score version II; SD, Standard Deviation; SOFA score, Sequential Organ Failure Assessment score; ICU, intensive care unit;. [a]The SAPS II was determined 24 h after ICU admission. [b]SOFA (Sequential Organ Failure Assessment) scores can range from 0 (no organ failure) to 24 (most severe level of multi-organ failure). The SOFA score is the sum of the six individual organ-failure sub-scores. Each sub-score can range from 0 (no organ failure) to 4 (most severe level of organ failure). Organ failure was defined as a score ≥2 on the relevant sub-scale.

## Treatments and outcomes

Vasoactive drugs and non-invasive ventilation were used more often in mainland France than in Réunion (Table 2). On the opposite, more patients in Réunion than on the mainland received renal replacement therapy.

ICU and hospital stay lengths, ICU mortality, and hospital mortality were not significantly different between the two groups (Table 2).

## Clinical phenotypes

MCA and HAC identified three patient clusters based on organ-failure phenotypes (Figs 1 and S2–S4 and Table 3). Cluster 1, the most common phenotype (n = 156; 54.5%), was character-ised by severe hepatic, renal, and haematologic failure (relevant SOFA sub-scores ≥2) con-trasting with the absence of neurological failure. This pattern matched the description of Weil's disease. The mortality rate in cluster 1 was 8.3% (13/156). Cluster 2 (n = 110; 38.5%) was composed of the patients who had the least severe critical illness and the lowest mortality rate, of 1.8%. The prevalence of renal and haematological failure was lowest in this cluster, which shared with cluster 1 an absence of patients with neurological failure. Cluster 3 was the rarest phenotype (n = 20; 7%) but also the most severe, with all patients exhibiting neurological failure and most having respiratory failure (85%) and cardiovascular failure (90%). This cluster

**Table 2. Intensive care unit treatments and outcomes.**

| | Total N = 288 | Mainland France N = 160 | Réunion N = 128 | *p* |
|---|---|---|---|---|
| **Organ support** | | | | |
| Vasoactive drugs, n (%) | 133 (46.2) | 92 (57.5) | 41 (32) | <0.001 |
| Time on vasoactive drugs, days, mean±SD | 4.2 ± 3.6 | 4.5 ± 4.0 | 3.7 ± 2.4 | 0.20 |
| Invasive mechanical ventilation | 96 (33.3) | 57 (35.6) | 39 (30.5) | 0.38 |
| Time on invasive mechanical ventilation, days, mean±SD | 9.9 ± 8.9 | 10.5 ± 10.8 | 10 ± 4.9 | 0.35 |
| Renal replacement therapy | 129 (44.8) | 56 (35) | 73 (57) | <0.001 |
| Time on renal replacement therapy, days, mean±SD | 7.2 ± 7.2 | 8.5 ± 8.6 | 6.2 ± 5.7 | 0.08 |
| Non-invasive mechanical ventilation, n (%) | 44 (15.3) | 32 (20) | 12 (9.4) | 0.03 |
| Extracorporeal membrane oxygenation, n (%) | 8 (2.8) | 3 (1.9) | 5 (4) | 0.47 |
| **ICU stay length, days, mean±SD** | 8.0 ± 8.7 | 8.2 ± 10.3 | 7.6 ± 6.2 | 0.50 |
| **Hospital stay length, days, mean±SD** | 15.2 ± 13 | 16.1 ± 14.6 | 14.1 ± 10.4 | 0.18 |
| **In-ICU death** | 21 (7.3) | 13 (8.1) | 8 (6.2) | 0.65 |
| **In-hospital death** | 22 (7.6) | 14 (8.8) | 8 (6.2) | 0.51 |

ICU: intensive care unit

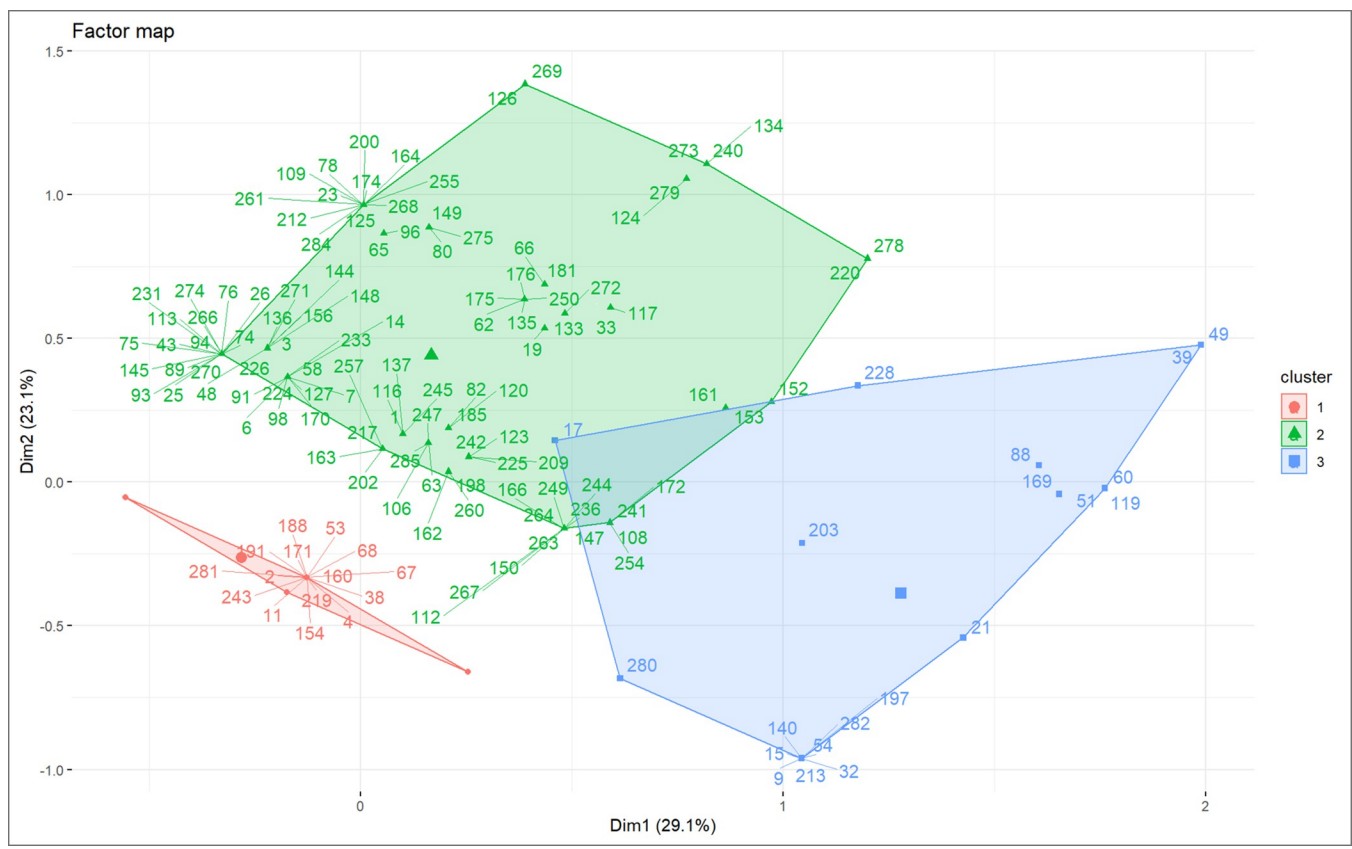

**Fig 1. Map of clinical features of severe leptospirosis.** Two-dimensional distribution of clinical features of severe leptospirosis mapped along the first two dimensions (Dim 1 and Dim 2) identified by multiple correspondence analysis. These dimensions summarised 52.2% of the variability in the data. Hierarchical ascending classification showed that the patients were distributed in the plane in three clusters reflecting three distinct clinical phenotypes of severe leptospirosis. Cluster 1 (black; n = 167) was characterised by severe hepatic, renal, and coagulation failure; cluster 2 (red; n = 73) by moderately severe disease; and cluster 3 (green; n = 14) by severe neurological failure, usually with severe respiratory and cardiovascular failure.

had the longest ICU and hospital stay lengths and the highest mortality rate (30%). The proportions of clinical phenotypes did not differ between geographical area (p = 0.1).

## Discussion

This large multicentre retrospective study provides the first comparison of severe leptospirosis in a tropical region (Réunion) *vs*. a temperate region (mainland France) with similar healthcare access, practices, and resources [10,11]. Mean age was younger in Réunion, which also had higher proportions of patients with myalgia, jaundice, and haemoptysis. In the overall population, we identified three clinical phenotypes: hepato-renal-haematological, moderately severe, and neurological, with mortality rates of 8.3%, 1.8%, and 30%, respectively. None of the patients with the first two phenotypes had neurological failure. Importantly, no significant differences between the Réunion and mainland cohorts were found for ICU or hospital stay length or for ICU or hospital mortality rate.

The patients were closely similar in the Réunion and mainland cohorts. In particular, over 90% of patients were men in both regions, in keeping with the higher proportion of men than women who engage in activities associated with *Leptospira* exposure. The slightly younger age in Réunion may reflect the younger mean age on the island compared to mainland France. Comorbidities were present in similarly low proportions of patients in both regions. Patients

**Table 3. Comparison of the three patient clusters identified by hierarchical ascending classification.**

| Characteristic | Total<br>N = 286 | Cluster 1<br>N = 156 | Cluster 2<br>N = 110 | Cluster 3<br>N = 20 | p |
|---|---|---|---|---|---|
| **Patients, n (%)** | | | | | 0.1 |
| Mainland France | 160 (55.9) | 83 (53.2) | 69 (62.7) | 8 (40) | |
| Réunion | 126 (44.1) | 73 (46.8) | 41 (37.3) | 12 (60) | |
| **Features on ICU admission** | | | | | |
| Age, years, mean±SD | 47 ± 16 | 48 ± 17 | 47 ± 15 | 46 ± 15 | 0.94 |
| Chronic alcohol abuse, n (%) | 62 (21.7) | 39 (25) | 14 (12.7) | 9 (45) | 0.002 |
| Smoking, n (%) | 94 (32.9) | 48 (30.8) | 38 (34.6) | 8 (40) | 0.63 |
| Fever, n (%) | 242 (84.6) | 133 (85.3) | 96 (87.3) | 13 (65) | 0.10 |
| Myalgia, n (%) | 196 (68.5) | 116 (74.4) | 72 (65.5) | 8 (40) | 0.005 |
| Arthralgia, n (%) | 76 (26.6) | 38 (24.4) | 35 (31.8) | 3 (15) | 0.19 |
| Abdominal pain, n (%) | 85 (29.7) | 52 (33.3) | 26 (23.6) | 7 (35) | 0.20 |
| Dyspnoea, n (%) | 75 (26.2) | 31 (19.9) | 33 (30) | 11 (55) | 0.002 |
| Cough, n (%) | 80 (28) | 37 (23.7) | 38 (34.5) | 5 (25) | 0.15 |
| Haemoptysis, n (%) | 51 (17.8) | 23 (14.7) | 20 (18.2) | 8 (40) | 0.02 |
| **SOFA sub-score $\geq 2^a$, n (%)** | | | | | |
| Central nervous system failure | 20 (7) | 0 (0) | 0 (0) | 20 (100) | <0.001 |
| Renal failure | 214 (74.8) | 156 (100) | 44 (40) | 14 (70) | <0.001 |
| Cardiovascular failure | 116 (40.6) | 59 (37.8) | 39 (35.5) | 18 (90) | <0.001 |
| Liver failure | 224 (78.3) | 156 (100) | 55 (50) | 13 (65) | <0.001 |
| Coagulation failure | 239 (83.6) | 156 (100) | 71 (64.5) | 12 (60) | <0.001 |
| Respiratory failure | 102 (35.7) | 47 (30.1) | 38 (34.5) | 17 (85) | <0.001 |
| **Life support, n (%)** | | | | | |
| Non-invasive ventilation | 43 (15) | 22 (14.1) | 16 (14.5) | 5 (25) | 0.44 |
| Invasive mechanical ventilation | 95 (33.2) | 54 (34.6) | 22 (20) | 19 (95) | <0.001 |
| Vasoactive drugs | 132 (46.2) | 68 (43) | 45 (37) | 19 (93) | <0.001 |
| Renal replacement therapy | 127 (44.4) | 97 (62.2) | 17 (15.5) | 13 (65) | <0.001 |
| **Outcomes** | | | | | |
| Death in the ICU, n (%) | 21 (7.3) | 13 (8.3) | 2 (1.8) | 6 (30) | <0.001 |
| Death in the hospital, n (%) | 22 (7.7) | 13 (8.3) | 2 (1.8) | 7 (35) | <0.001 |
| ICU stay length, days, mean±SD | 8 ± 8.7 | 9.4 ± 10 | 5 ± 4.3 | 12.8 ± 11.9 | <0.001 |
| Hospital stay length, days, mean±SD | 15.2 ± 13 | 16.9 ±13.7 | 10.6 ± 6.5 | 27.5 ± 22 | <0.001 |

ICU, intensive care unit; SOFA score, Sequential Organ Failure Assessment score.

with comorbidities are less likely than are healthy individuals to engage in the occupational and recreational activities associated with exposure to *Leptospira*.

Importantly, the clinical features of severe leptospirosis did not differ between the tropical and temperate regions. MAC identified three phenotypes characterised by differences in target organs and mortality. The neurological phenotype was both the least common and the most severe, with nearly a third of patients dying. Neurological failure occurred only in this phenotype, usually in combination with cardiovascular and respiratory failure. The most common form manifested as hepatorenal failure and thrombocytopenia, or Weil's disease. This form was of intermediate severity. Finally, the least severe phenotype, found in slightly more than a third of patients, was rarely associated with renal or coagulation failure. None of these phenotypes differed in frequency between the two regions. However, there were more frequent cardiovascular failures on admission in the mainland cohort than in Reunion.

The severity scores on ICU admission were high in both regions, predicting a poor prognosis. Overall, 46.2%, 33.3%, and 44.8% of patients required vasoactive drugs, invasive mechanical ventilation, and renal replacement therapy, respectively. However, the overall 7.3% hospital mortality rate was far lower than expected based on the severity scores and was also lower than in previous studies of severe leptospirosis or of other infections responsible for multi-organ failure [15,21,22]. ICU and hospital stay lengths were also shorter than predicted by the severity scores, with no significant difference between the two regions. In previous studies, mortality rates among adult ICU patients with severe leptospirosis ranged from 4% in tropical Australia to 52% in India [6,7,15,21–23]. In a tropical part of Australia where healthcare was similar to that in France, the median SAPS II was 32 [17–55] indicating somewhat less severe disease than in our population, mechanical ventilation and renal replacement therapy were required by 49% and 33% of patients, and mortality was 4% [6]. Thus, mortality may be low when high-quality critical-care resources are available, regardless of whether severe leptospirosis occurs in a tropical or temperate region.

One limitation of our study is the combination of data from two different studies. This design required us to limit the characterisation of clinical patterns to organ failures, for which the definitions were identical in the two cohorts, being based on SOFA sub-score values. The recruitment periods differed between the two studies, although not to the extent that an impact on patient features, treatments, or outcomes would be expected. Second, both of the studies collected the data retrospectively, indicating a risk of data collection bias and missing information. Patients in Réunion were admitted over 11 years, a period that would not seem long enough to result in major changes in the management of severe leptospirosis or in the distribution of causative *Leptospira* species. The mainland cohort was established over only 4½ years. Third, *Leptospira* species identification was performed only rarely in mainland France and not at all in Réunion. However, previous data indicate that the most common species are the same in the two regions [24,25]. The main strength and originality of our study is the comparison of two regions that shared the same healthcare culture and resources but had markedly different climates. The similarity in severity scores reflects the similarity in ICU-admission criteria between Réunion and mainland France. Also, the comparable times from symptom onset to admission indicate comparable access to care.

## Conclusions

This large multicentre retrospective study of severe leptospirosis requiring ICU admission showed limited differences in patient features and clinical patterns between a tropical region and a temperate region of France that had similar healthcare access, practices, and resources. Outcomes did not differ between these two regions. We identified three clinical phenotypes, which may assist in the early diagnosis and management of severe leptospirosis worldwide.

## Supporting information

**S1 Text. Supplemental File.**
(DOCX)

**S1 Fig. Seasonal distribution of severe leptospirosis in Réunion and mainland France.**
Number of cases of severe leptospirosis, per month, in both cohorts.
(TIF)

**S2 Fig. Percentage of inertia returned by each dimension of the multiple correspondence analysis.** Two patients from Réunion were excluded from the clinical-phenotype analysis due to missing data on organ failure as defined by the SOFA sub-scores. This left 286 patients for

the analysis. By multiple correspondence analysis, the cumulative inertia was 66.9% for the first three dimensions. Inertia was 29.1% for the first dimension and 23.1% for the second dimension, yielding a cumulative inertia of 52.2% for the first two dimensions.
(TIF)

**S3 Fig. Relationship between the first and second dimensions for the variables, with death in the ICU and cohort of inclusion as illustrative variables.** In the first dimension, central nervous system failure, cardiovascular failure, and respiratory failure (righthand side of the graph) are diagonally opposite absence of these events (lefthand side of the graph). Note the close proximity of the two cohorts. CNS: central nervous system; CV: cardiovascular; coag: coagulation; respi: respiratory
(TIF)

**S4 Fig. Cluster dendrogram produced by hierarchical clustering on the principal components.** A dendrogram is a tree diagram that illustrates the arrangement of the clusters. Here, individuals are plotted on the X axis. The green, black, and red rectangles define the three clusters of patients: hepato-renal leptospirosis, moderately severe leptospirosis, and neurological leptospirosis, respectively. The graph in the upper right corner represents the loss of inertia along the different dimensions.
(TIF)

**S1 Acknowledgments. The list of contributors of the study.**
(DOCX)

## Acknowledgments

We are indebted to A. Wolfe, MD, who helped to prepare and review the manuscript. The list of contributors is provided in S1 Acknowledgments.

## Author Contributions

**Conceptualization:** Jérôme Allyn, Lucas Marti, Nicolas Allou, Jean Reignier.

**Data curation:** Arnaud-Félix Miailhe, Benjamin Delmas, Julien Jabot.

**Formal analysis:** Jérôme Allyn, Benjamin Delmas, Lucas Marti, Jean Reignier.

**Investigation:** Jérôme Allyn, Arnaud-Félix Miailhe, Benjamin Delmas, Julien Jabot, Jean Reignier.

**Methodology:** Jérôme Allyn, Arnaud-Félix Miailhe, Lucas Marti, Nicolas Allou, Julien Jabot, Jean Reignier.

**Resources:** Nicolas Allou.

**Software:** Jérôme Allyn, Nicolas Allou.

**Supervision:** Jérôme Allyn, Julien Jabot, Jean Reignier.

**Validation:** Jérôme Allyn, Arnaud-Félix Miailhe, Benjamin Delmas, Nicolas Allou.

**Visualization:** Jérôme Allyn.

**Writing – original draft:** Jérôme Allyn, Lucas Marti.

**Writing – review & editing:** Jérôme Allyn, Benjamin Delmas, Nicolas Allou, Julien Jabot, Jean Reignier.

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
