## [Decision Letter · Decision Letter 0]

19 Mar 2024

Dear Dr. Allyn,

We are pleased to inform you that your manuscript 'Severe Leptospirosis in Tropical and Non-Tropical Areas: A Comparison of Two French, Multicentre, Retrospective Cohorts' has been provisionally accepted for publication in PLOS Neglected Tropical Diseases.

On behalf of the journal, I apologize for the length of time required to complete this review. We contacted a large number of potential reviewers before one agreed to assist us. I also reviewed the manuscript, and concur with the outside reviewer.

Best regards,

Brian Stevenson, Ph.D.

Academic Editor

Ana LTO Nascimento

Section Editor

Reviewer's Responses to Questions

**Key Review Criteria Required for Acceptance?**

**Methods**

-Are the objectives of the study clearly articulated with a clear testable hypothesis stated?

-Is the study design appropriate to address the stated objectives?

-Is the population clearly described and appropriate for the hypothesis being tested?

-Is the sample size sufficient to ensure adequate power to address the hypothesis being tested?

-Were correct statistical analysis used to support conclusions?

-Are there concerns about ethical or regulatory requirements being met?

Reviewer #1: The objectives of the study are clearly articulated with a clear testable hypothesis stated.

The study design is appropriate to address the stated objectives?

The population is clearly described and appropriate for the hypothesis being tested?

The sample size is sufficient to ensure adequate power to address the hypothesis being tested?

Statistical analysis were correct

No concerns about ethical or regulatory requirements being met?

**Results**

-Does the analysis presented match the analysis plan?

-Are the results clearly and completely presented?

-Are the figures (Tables, Images) of sufficient quality for clarity?

Reviewer #1: The analysis presented match the analysis plan

The results are clearly and completely presented

Tables are of sufficient quality for clarity

**Conclusions**

-Are the conclusions supported by the data presented?

-Are the limitations of analysis clearly described?

-Do the authors discuss how these data can be helpful to advance our understanding of the topic under study?

-Is public health relevance addressed?

Reviewer #1: The conclusions are supported by the data presented

The limitations of analysis are clearly described?

The authors discuss how these data can be helpful to advance our understanding of the topic under study

**Editorial and Data Presentation Modifications?**

Reviewer #1: (No Response)

**Summary and General Comments**

Reviewer #1: Authors compared two retrospective cohorts of patients admitted to intensive care for severe leptospirosis, one from Reunion Island in the Indian Ocean (tropical climate) and the other from metropolitan France (temperate climate). After grouping the two cohorts, authors also performed multiple correspondence analysis and hierarchical clustering to search for distinct clinical phenotypes. The Réunion and Metropolitan France cohorts comprised 128 and 160 patients respectively. Compared with the Réunion cohort, the metropolitan cohort had a higher mean age (42.5±14.1 vs. 51.4±16.5 years, p<0.001). Severity scores, length of stay and mortality did not differ between the two cohorts. Three phenotypes were identified: hepato-renal leptospirosis (54.5%) characterized by significant hepatic, renal and coagulation failure, with a mortality of 8.3%; moderately severe leptospirosis (38.5%) with less severe organ failure and the lowest mortality rate (1.8%); and very severe leptospirosis (7%) manifested by neurological, respiratory and cardiovascular failure, with a mortality of 30%.

This is an interesting study, based on a large number of patients and a robust methodology. The authors find no clinically significant difference between the two cohorts, apart from the age of the patients (younger on Reunion Island). The description of the three patient phenotypes is very interesting. It corresponds well to what is observed in practice and is useful for patient triage and management.

It would be interesting, if possible, to supplement these clinical descriptions with data on exposure to the risk of leptospirosis and bacteriological data (serotype and genotype). Differences could be observed between the two cohorts and explain certain differences.

The data presented in this study complements those already known about severe forms of leptospirosis, and deserves to be published.

PLOS authors have the option to publish the peer review history of their article (what does this mean?). If published, this will include your full peer review and any attached files.

Reviewer #1: **Yes: **André Cabié

---

## [Editor Report · Acceptance letter]

2 Apr 2024

Dear Dr Allyn,

We are delighted to inform you that your manuscript, "Severe Leptospirosis in Tropical and Non-Tropical Areas: A Comparison of Two French, Multicentre, Retrospective Cohorts," has been formally accepted for publication in PLOS Neglected Tropical Diseases.

Best regards,

Shaden Kamhawi

co-Editor-in-Chief

Paul Brindley

co-Editor-in-Chief
